# Assessing the influence of diverse skills on employability outcomes for IT undergraduates

**Dhamindi Senadheera[1], Krishantha Wisenthige [2]***

1 Faculty of Graduate Studies, Sri Lanka Institute of Information Technology, Malabe, Sri Lanka,
2 Department of Business Management, SLIIT Business School, Sri Lanka Institute of Information Technology, Malabe, Sri Lanka

* krishantha.w@sliit.lk

## Abstract

Rapid technological advancements have reshaped the global job market, emphasizing the importance of specialized competencies such as user interface (UI) and user experience (UX) design, alongside technical and interpersonal skills.. This study examines how UI/UX skills (UIUX), soft skills (SS), and technical skills (TS) influence the employability (EP) of IT undergraduates in Sri Lanka, addressing a notable gap in existing literature that often examines these competencies in isolation and predominantly within Western contexts. The current study offers a detailed examination of employability determinants in Sri Lanka's IT sector by incorporating gender as a moderating factor and investigating the mediating roles of self-efficacy (SE) and proficiency levels (LP). The collection of data involved 345 IT undergraduates participating in structured surveys, which were subsequently analyzed using partial least squares structural equation modelling (PLS-SEM). The results demonstrate that gender significantly affects the relationship between soft skills and technical skills with employability, underscoring differences in the assessment of these competencies among different genders. Moreover, the degree of proficiency influences the connection between technical skills and employability, yet it does not play a significant mediating role in the relationship between soft skills and UI/UX employability. Self-efficacy has proven to be a significant mediator across various skill categories UI/UX, soft, and technical highlighting its essential function in converting competencies into career success. This work seeks to add to existing knowledge by tackling the main significant gap of examining the combined effect of UI/UX, soft, and technical skills on employability. This study contributes to the theoretical understanding of employability by presenting an integrated model that elucidates the complex interactions among skills, mediators, and gender within the Sri Lankan IT sector. The results provide actionable insights for educators, policymakers, and industry leaders, advocating for curriculum alignment with industry needs and the promotion of self-efficacy through mentorship and experiential learning.

**Data availability statement:** All relevant data are within the manuscript and its Supporting Information files.

**Funding:** The author(s) received no specific funding for this work.

## 1. Introduction

Rapid technological advancements are transforming industries worldwide, fundamentally reshaping the global labor market and driving a significant increase in demand for specialized roles like user interface (UI) and user experience (UX) designers. This shift grants both significant opportunities and challenges for higher education institutions requiring them to ensure graduates hold the essential skills for success in these emerging fields [1]. The designed model proposes determining if IT undergraduate skills affect UI/UX design jobs.

IT degrees are in demand in Sri Lanka, reflecting the global move toward digitalization and the expansion of technology-driven businesses [2]. Employability is "the ongoing fulfilment, acquisition, or creation of work through the optimal utilization of competencies" [2]. In addition, communication, problem-solving, and critical thinking are crucial for IT success, according to studies [2]. Employers value practical knowledge, analytical skills, job dedication, communication and IT skills, management acumen, and favorable attitudes [3]. Listening, interpersonal, verbal and written communication, critical thinking, professionalism, creativity, adaptability, professional confidence, job-specific competencies, leadership, and work experience are also important [3]. However, most existing studies examine UI/UX, technical, and soft skills as individual, and are primarily focused on Western contexts. This leaves a significant gap in understanding how these competencies collectively influence employability in Sri Lanka's IT sector. Research is required to evaluate the correlation between the competencies of IT students and the specific requirements of UI/UX design positions.

The conducted study examines the intersection of UI/UX capabilities, soft skills, and technical skills in Sri Lanka and graduate preparation for UI/UX careers to address gaps in the literature. This study aims to address these gaps by investigating how UI/UX skills, soft skills, and technical skills among Sri Lankan IT undergraduates impact their employability. It also examines the mediating roles of proficiency and self-efficacy, as well as the moderating effect of gender, in shaping career outcomes. This research effort addresses: what is the impact of UI/UX skills, soft skills and technical skills of Sri Lankan undergraduates for employability? The primary objective is to investigate the impact of UI/UX skills, soft skills and technical skills of Sri Lankan undergraduates for employability. Sri Lanka may capitalize on the global need for UI/UX professionals by training a skilled workforce. This will provide valuable insights for institutions seeking to improve their IT education programs. This effort will educate IT undergraduates about UI/UX capabilities, helping them make educational and career decisions aiming to improve the employability of IT graduates in Sri Lanka's expanding UI/UX design sector by addressing existing gaps. This research delineates the precise objectives it seeks to achieve among Sri Lankan undergraduates from both public and private universities. These objectives are closely linked to research issues and direct the study process.

This research is anticipated to yield many substantial contributions that might greatly influence skill sets and inform future practices in undergraduate UI/UX employability. Initially, it will offer an in-depth comprehension of the essential skills and characteristics required for IT graduates to excel in UI/UX design positions within

the Sri Lankan environment. Secondly, by examining the mediating roles of proficiency and self-efficacy, this research will provide insights into the fundamental mechanisms that facilitate professional success in this domain. Third, by investigating the potential moderating influence of gender, the study will enhance the comprehension of gender dynamics and any inequities within the UI/UX design industry. This research will provide significant data-driven suggestions for stakeholders, including students, educators, and policymakers, aiming to improve IT graduate employability in Sri Lanka's expanding UI/UX design sector.

The following subsections are organized as follows. Literature review section offers an extensive literature survey, integrating significant academic contributions that underpin the theoretical framework of this study. Methodology section outlines the technique utilized, encompassing the study design, data gathering, and analysis processes. Results section presents the findings derived from the analysis. Discussion section presents an analysis of this data, clarifying their significance in relation to the current literature. Implication for IT education in Sri Lanka section examines the implications of the findings explicitly for IT education. Limitations of the study and Suggestions for future research sections outline the study's shortcomings and provide prospects for further research. The paper concludes with Conclusion section, which provides a summary of the primary findings and their broader implications.

## 2. Literature review

The rapid evolution of technology has redefined job roles and skill requirements across various industries, placing particular emphasis on user interface (UI) and user experience (UX) design skills. In an increasingly digital landscape, the alignment between higher education curricula and industry demands is critical for enhancing the employability of graduates in the IT sector [4,5]. This literature review seeks to explore the existing research on the impacts of UI/UX skills (UIUX), technical skills (TS), and soft skills (SS) on the employability (EP) of IT undergraduates, particularly in the context of Sri Lanka. The user-centered design's role in customer satisfaction and company success has boosted demand for professional UI/UX designers and as companies adapt to digital change, they prefer personnel with talents that enable smooth and engaging user experiences because user experience is crucial to businesses' performance in today's competitive environment, creating need for qualified professionals in this industry [1,2].

Research in Sri Lanka highlights a critical need to align educational outcomes with industry demands, thereby boosting graduate employability within the expanding IT sector. According to Samarasinghe (2022) [2] the local education system must adapt to the changing demands posed by technological advancements to ensure that graduates possess the requisite UI/UX design, technical, and soft skills. Existing research by Grosemans (2024) [6] a skills gap analysis reveals that although the number of IT graduates continues to rise, many lack practical competencies that meet employer expectations. Studies emphasize the importance of integrating industry-relevant training and curricular reforms to bridge these gaps [7]. To address the critical skills gap in Sri Lanka, particularly in the UI/UX domain, stakeholders including educational institutions, policymakers, and industry leaders must engage in collaborative efforts to enhance the skillsets of IT graduates. Programs that promote both technical skills training and self-efficacy development through mentorship and targeted workshops are vital for creating a capable workforce [8–10].

### Employability and the growing landscape of skill sets

The notion of employability was originally examined through a macroeconomic lens, concentrating on governmental strategies aimed at improving employment rates. However, there has been a notable shift towards a focus on the individual [11,12]. Employability is increasingly acknowledged as a fluid characteristic, reflecting a graduate's ability to not only obtain initial employment but also to maintain and progress in their career path [6,13–16]. This capability is supported by a blend of accomplishments, encompassing skills, knowledge, and personal qualities, that increase the chances of secure and thriving in selected professions [11,17,18]. The focus is not solely on having a degree; it is about the comprehensive array of skills that empower individuals to adeptly move through the difficulties of the job market [18–23].

 

Perceived employability refers to an individual's confidence in their ability to secure and retain employment in a competitive and evolving labor market [24,25]. This confidence is a complex psychological construct shaped by various factors, rather than merely an optimistic perspective. It encompasses an individual's self-assurance in their competencies, expertise, and skills, along with their understanding of how these attributes align with current and future labor market requirements [26,27]. Employability research operates across three conceptual strands: personal strengths that increase employment potential, self-perceived employment opportunities, and actual job transitions as a realization of that potential [28]. Critically, scholars advocating the self-perceived employment approach argue that the personal strengths that increase employment conflict with its origin, therefore treating skills and confidence as synonymous with perceived employability commits a category error [28]. In the current dynamic economic landscape, characterized by continual industry evolution and the emergence of new roles, confidence in one's job-seeking abilities is a significant predictor of successfully obtaining and retaining employment. While confidence is an internal, affective state, perceived employability is an external, outcome-oriented judgment that incorporates feedback from recruiters, labor-market signals, and personal constraints [29]. Consequently, interventions aiming to enhance graduates' employability must address not only skill development and self-confidence but also market awareness, networking opportunities, and realistic appraisal of occupational demands [30,31].

Employability is a multi-dimensional construct involving mutual interactions between individual favor and organizational cues, highlighting how interpretation of key events informs individuals and their shapes career paths [32]. Under this framework, perceived employability is negotiated in context that it is coproduced through encounters with labor markets, employers, and professional communities [32]. Findings support the conclusion reached by Soares and Mosquera that perceived employability is related to, but distinct and much more than, employability skills and attributes [33,34].

Research has consistently recognized the importance of a combination of technical, soft, and personal skills for enhancing the employability of IT graduates [2,10,19–22,35–37]. Specifically, technical skills in technology deployment and management are crucial for graduates to meet the rigorous demands expected by employers [20,36,38,39]. Studies have noted that graduates must not only possess solid technical knowledge but also exhibit essential soft skills such as critical thinking, adaptability, and effective communication [3,21,22,35]. As noted by Hartson (2019) [1], the effectiveness of digital products hinges significantly on user-centered design principles. The requirement for capable professionals in this field is underscored by the rapid integration of digital platforms into various industries, creating a pressing need for graduates who can bridge the gap between technology and user experience [2,40,41]. UI/UX experts must be technically proficient and creative problem-solvers to be employable. In recent years, shifting labor market needs and graduate unemployment worries have drawn focus to employability [40,42]. For IT undergraduates, perceived employability is shaped by three interrelated factors: Technical competencies and soft skills provide the foundation for self-assessment, yet the relevance of these skills to current industry demand determines how graduates evaluate their prospects [43]. Networks, mentorship, and internship experiences grant access to hidden opportunities, thereby enhancing perceived employability beyond what skills alone can explain a high level of perceived employability is linked to greater career planning behavior, proactive job search activities, and lower academic anxiety [30,44]. Conversely, low perceived employability can undermine motivation, increase dropout risk, and exacerbate stress [45].

This research analyses the current research on employability, concentrating on students and IT graduates. It examines many definitions and frameworks of employability, determinants affecting graduate employability, and the contribution of higher education institutions in cultivating employable graduates and their skills.

Yorke and Knight emphasize the complexities of employability by underscoring the importance of skills, self-efficacy beliefs, metacognition, comprehension, and learning [46]. Employability metrics encompass perceived employability, job performance, and occupational success [47]. Harvey's research underscores the significance of higher education in fostering attitudes conducive to sustained employability and career progression, while also aiding students in acquiring

transferable skills [48]. Employability influences not only initial recruitment but also long-term results, including employee retention and engagement. Outcomes that improve IT undergraduates' market awareness, networking opportunities, and realistic skill to job mapping are essential for fostering positive employability perceptions and, consequently, better academic and career outcomes [49].

Ultimately, employers are essential as they invest in graduate training programs, provide honest feedback on skill deficiencies, and actively strive to optimize students' potential. Employability is authentically defined and cultivated within this collaborative ecosystem, where employers, institutions, and students are all actively engaged [50].

The sections that follow examine deeper into the significance and interaction of UIUX skills, soft skills, and technical skills concerning graduate employability. The subsequent section will outline the development of the study's hypotheses.

### The direct impact of different skill domains on employability

**The influence of UI/UX skills on job marketability.** UI/UX design is an emerging career focused on enhancing user experience and happiness; nevertheless, consensus on the requisite skills remains elusive [6,39,51,52]. UI/UX design is an expanding profession that is gaining significance across all enterprises utilizing digital technologies. This concept originated from the integration of human-computer interaction, product design, interface design, communication design, business practices, and information technology [1,39,53]. Hassenzahl and Tractinsky (2006) [54] define "user experience" as encompassing usability, aesthetic appeal, functionality, and emotional response during technology usage. Ardito (2021) [55] assert that students acquire fundamental UX skills via short courses, workshops, and knowledge transfer prior to doing internships. Borriraklert and Kiattisin (2021) [39] emphasize that user experience knowledge includes design ideas, methodologies, and research approaches. The abilities encompassed design concepts, design processes, aesthetic value, information art, business acumen, and project management.

UI/UX professionals need many skills to create user-focused digital experiences and to understand user needs and behaviors, do user research, create user interfaces and interactions, prototype and test designs, and iterate depending on user input. It comprises understanding of design ideas, usability criteria, accessibility standards, and applicable applications and technology [40,51,56–59]. Bilousova and colleagues (2021) [51] emphasize hands-on UI and UX design training and they recommend courses on visual perception psychology, graphic interface design, and interface design tools. Patacsil and Tablatin (2017) [60] underscore the significance of both soft and hard skills for IT internships and the industry, indicating that UI/UX competencies are becoming progressively vital for entry-level roles. Scolere (2019) [61] examines the significance of portfolio development for designers, especially with social media and personal branding.

For IT undergraduates aiming for positions in UI/UX design, it is essential to acquire and showcase particular UI/UX skills to enhance their chances of employment in this competitive arena which involved include a variety of competencies focused on grasping user needs, crafting intuitive and effective interfaces, and guaranteeing a positive overall experience with digital products and services user research techniques, the capability to construct information architecture and user flows, knowledge of interaction design principles, a thorough understanding of visual design components, proficiency in creating interactive prototypes with industry-standard tools and the skill to perform usability testing and refine designs based on user feedback [39,40,51,58,61]. Including UI/UX skills into curricula enhances students' employability in the technology sector by augmenting their ability to address real-world challenges [58].

Furthermore, though the study explored skills related to employability, it did not go further into the specific nuances concerning the employability of Sri Lankan undergraduates. The presented analysis seeks to clarify the gaps found in earlier investigations. Consequently, considering all the preceding arguments, we propose the following for quantitative validation.

$H_1$ – UI/UX skills have a positive impact on the employability of IT undergraduates

## Soft Skills' significant effect on UI/UX employability

Soft skills encompass personal attributes and interpersonal abilities that facilitate effective collaboration and communication in the workplace. They are increasingly recognized as equally important as technical skills in ensuring job readiness for IT graduates [62]. Past research highlights that graduates often fall short in soft skills, leading to challenging transitions into the workforce and employers are prioritizing candidates who can navigate complex social environments and collaborate effectively with team members, underscoring the necessity for higher education institutions to integrate these skills into their curricula [6,60].

In Tanzania, a 2024 study demonstrated that undergraduates with stronger soft skills reported higher perceived employability yet emphasized that academic programs often prioritize technical training over interpersonal development [21]. Further research highlights emergent soft skills such as remote collaboration, self-directed learning, and stress management, which IT undergraduates in Sri Lanka lacked due to limited exposure to hybrid work simulations during their education [63].

Soft skills are personal qualities, psychological traits, and interpersonal abilities that facilitate effective interaction and collaboration in professional and social contexts [35]. Soft skills encompass communication, teamwork, problem-solving, critical thinking, time management, flexibility, leadership, and emotional intelligence [64,65]. Magogwe, Nkosana, and Ntereke emphasize that employers choose applicants who possess robust soft skills in addition to technical proficiency, since these abilities enhance communication, collaboration, and problem-solving in professional settings [66]. Pazil and Razak (2019) [67] underscore this by conducting a systematic analysis of Asian employers' views on graduates' soft skills, accentuating the necessity for adaptable graduates possessing these competencies. Soft skills facilitate individuals in managing workplace dynamics, fostering robust connections, and contributing successfully to team initiatives, so improving their employability and career opportunities [64].

UI/UX designers must proficiently connect with clients, comprehend user requirements, work with engineers, and convincingly show their designs [6]. Mechelen and other researchers (2019) [68] examine the evaluation of co-design competencies, encompassing creativity, empathy, and cooperation, which are vital for UI/UX experts. McKenzie, Coldwell-Neilson, and Palmer (2018) [69] examine the career development and employability of IT students, emphasizing the necessity of a comprehensive strategy that includes the cultivation of soft skills. This may entail integrating soft skills training into current curricula, providing specialized seminars, and facilitating chances for students to develop and implement these abilities in practical contexts [69].

According to past studies, students can enhance their communication skills, work well with others, solve problems, and adjust to the demands of the UI/UX design industry [60,70]. Consequently, there exists a significant opportunity to examine the intricate relationship between specific soft skills and their impact on the employability of IT undergraduates pursuing UI/UX positions. This study will provide significant insights for curriculum development and skill enhancement initiatives [71]. Studies conducted by Mata et al. (2021) [72] indicate that effective time management significantly reduces workplace stress and enhances individual work performance. Their findings indicate a strong positive correlation between job performance and time management skills, underscoring the significance of time management as a crucial soft skill in the workplace, especially in high-pressure scenarios. Essential soft skills that are particularly important for UI/UX professionals encompass effective communication, teamwork and collaboration, problem-solving, critical thinking, and presentation skills [21,22]. A study examining the fundamental skills for UI/UX designers revealed that communication and collaboration were deemed highly important by industry professionals, frequently regarded as critical as technical design skills [22]. Interpersonal abilities can serve as a crucial distinguishing factor for individuals with comparable technical skills, frequently impacting recruitment choices and playing a significant role in sustained career advancement within the sector.

Consequently, the research proposes the following.

$H_2$ —Soft skills have a positive impact on the employability of IT undergraduates.

 

## Technical skills' influence on job marketability

Technical skills are the specialized knowledge and competencies necessary to execute activities and employ tools within a certain domain [59]. In UI/UX, technical skills include knowledge in design software, prototype tools, user research techniques, and accessibility standards [40,51,58,59,73]. Potter (2020) [74] emphasizes the significance of technical and professional abilities for IT students engaged in project preparation, underscoring the necessity of integrating both skill sets. Further they discuss, technical expertise is frequently a prerequisite for entry-level roles and is essential for professional progression [74]. Sehgal and Nasim (2017) [75] examine the determinants of graduate employability in the Indian IT sector, emphasizing the importance of technical skills in a competitive labor market. A robust foundation in technical skills can markedly improve an individual's employability and facilitate access to many professional options labors market.

The discipline of UI/UX design is expanding swiftly, with a rising need for proficient people with technical skills and Rose (2020) [59] finds that the undergraduates aspiring to UI/UX employment must cultivate a robust foundation in domain-specific technical abilities. Jayasingha and Suraweera (2020) [3] analyze the determinants influencing graduate employability, emphasizing the significance of technical skills development in conjunction with academic knowledge. The relationship between technical skills and UI/UX design capabilities is essential for successful collaboration and the development of practical and efficient user interfaces. This comprehension enhances communication and collaboration with developers, resulting in a more streamlined and effective development process [76]. Technical skills are increasingly essential for employment in positions related to digital transformation and governance. Ali et al. (2021) [77] assert that robust IT governance frameworks necessitate technical personnel equipped for innovation, highlighting the significance of technical skill sets in both public and private sector employment. Their findings underscore the imperative for continuous IT skill enhancement to meet evolving organizational and digital innovation requirements.

The significance of technical skills in UI/UX is also consistent with wider digital competence frameworks [39]. This blend of design and technical skills enhances their appeal to employers looking for versatile professionals capable of contributing to multiple phases of the product development lifecycle.

To address this research gap, the present study hypothesis that,

$H_3$ —Technical skills have a positive impact on the employability of IT undergraduates.

## Moderating the relationship between skill and employability

**Moderating effect of gender.** The intersection of gender and skill development presents important considerations in understanding employability outcomes. Existing research indicates that gender may influence skill acquisition and the way skills are perceived by employers [78]. Noteworthy gender disparities exist in technical and soft skills development, affecting how male and female candidates approach job opportunities in the IT sector. The intersection of gender and skill acquisition remains pivotal in understanding employability outcomes among IT graduates. Several studies highlight that gender differences can influence self-efficacy and skill levels in technical fields, with male students often exhibiting higher self-efficacy and engagement in STEM disciplines compared to their female counterparts [79–81].

The role of gender can affect the intensity or orientation of the connection between skills, whether soft or technical, and employability [82]. This suggests that the influence of a specific level of skill proficiency on employability may vary between male and female IT undergraduates, influenced by factors such as societal expectations, gender stereotypes in the tech industry, differences in networking behaviors, or variations in the perception and valuation of skills based on gender [78,83]. The conceptual foundation for gender as a moderating factor arises from the recognition that social and cultural influences can profoundly impact career paths and opportunities [84]. Research indicates that gender and institutional type serve as moderators influencing the relationship between skills and employability [85–88]. The type of institution affects resource accessibility and teaching methodologies that influence proficiency development, while documented gender disparities in self-efficacy and skill application within technology education affect career outcomes. The integration of these moderators into the model facilitates a more nuanced comprehension of employability pathways.

 

More findings indicate that gender may influence the connection between soft skills and employability for IT undergraduates [89,90]. Some studies suggest that specific soft skills, like empathy or communication, may be viewed differently or held in higher regard in candidates of a certain gender, though these results tend to be complex and vary by context. It is plausible that hiring biases, both conscious and unconscious, influences the way in which the soft skills on employability differ according to gender.

Consequently, considering the evidence presented, the study posits the following hypothesis.

H₄ – Gender has a significant moderating effect on the relationship between soft skills and employability of IT undergraduates.

Studies have also examined how gender may influence the connection between technical skills and employability for IT undergraduates [82,83]. Even with initiatives aimed at enhancing gender diversity in the tech sector, prevailing gender stereotypes concerning technical skills may continue to shape perceptions and impact hiring choices [84].

Possible reasons for this moderation may include underlying biases concerning technical abilities, where male candidates are often viewed as more technically skilled or assertive, despite having comparable proficiency to their female peers [83,91,92]. It is essential for educators and employers to recognize these potential biases and adopt strategies to reduce their effects. This ensures that all IT undergraduates are afforded equal opportunities based on their technical skills and capabilities [77].

Consequently, considering the evidence presented, the study posits the following hypothesis.

H₅ – Gender has a significant moderate effect on the relationship between technical skills and employability of IT undergraduates.

## Mediating the relationship between skill and employability

**Mediating impact of level of proficiency.** Proficiency levels, particularly in relation to UI/UX skills and technical capabilities, significantly affect the employability of IT graduates. Studies have shown that higher proficiency levels in these skill sets correlate positively with job placement rates and satisfaction within tech roles [73]. The framework developed by Dacre Pool and Sewell (2013) [93] indicates that employability is contingent upon both hard and soft skills training programs must address the nuances of skill proficiency to foster successful employment outcomes. A study conducted by Almaiah and Al-Khalifa [94] highlights that practical training experiences increase student proficiency in technical domains, suggesting that structured curricula that incorporate hands-on learning effectively prepare graduates for industry demands

The proficiency level of an IT undergraduate, in particular UI/UX and technical skills serves as a crucial mediator in the connection between possessing these skills and attaining employability [39]. Having just a fundamental grasp of a skill might not be enough to land a competitive position; a level of proficiency is typically necessary to showcase value to prospective employers [94,95].

Recent studies indicate that the level of proficiency in soft skills plays a crucial role in mediating their influence on employability for IT undergraduates [96,97]. Having fundamental communication or teamwork skills is advantageous, but achieving a more advanced level of proficiency in these domains frequently enhances employability in IT proficient [96]. Exhibiting a strong command of the essential soft skills pertinent to a specific UI/UX position can greatly improve an IT undergraduate's chances of securing employment [94].

Consequently, considering the evidence presented, the study posits the following hypothesis.

H₆ – Level of proficiency has a mediating impact on the relationship between soft skills and employability of IT undergraduates.

In a similar vein, the level of expertise in technical skills serves as a crucial intermediary factor in the connection between having these skills and attaining employability for IT undergraduates [94,98]. A foundational grasp of HTML or CSS can be advantageous, yet a more advanced skill set in front-end development, particularly with JavaScript frameworks or familiarity with design tools, is frequently essential to attain more competitive positions [94,99,100].

Employers frequently look for professionals who possess significant expertise in at least one critical technical domain pertinent to their position [88]. IT undergraduates can effectively illustrate their proficiency in technical skills by presenting a robust portfolio that highlights intricate projects, relevant technology certifications, or active participation in open-source initiatives [61]. This solid proof of technical expertise greatly improves their job prospects in the UI/UX domain [61].

Consequently, considering the evidence presented, the study posits the following hypothesis.

$H_7$ – Level of proficiency has a mediating impact on the relationship between technical skills and employability of IT undergraduates.

## Mediating impact of self-efficacy

Self-efficacy plays a significant role in graduates' confidence in performing specific tasks and in their overall employability. Recent studies suggest that individuals with high levels of self-efficacy are more likely to pursue career opportunities and demonstrate resilience in overcoming challenges in the job market [7–9,96,101]. Self-efficacy, defined as an individual's belief in their capability to execute tasks effectively, has been linked to enhanced performance across various academic and professional settings [93,102,103]. Research shows that self-efficacy captures only one dimension of the self-assessment process, namely the belief in one's capability to perform specific tasks, whereas perceived employability integrates expectations about demand for those skills, the relevance of one's qualifications to available positions, and the likelihood of being selected by employers [44,104]. The literature indicates that students with higher self-efficacy are more motivated to engage in challenging tasks and persist longer in the face of difficulties [102,105–107]. Research by Pak and other researchers illustrates that self-efficacy plays a critical role in enhancing the employability of graduates, demonstrating its function as a crucial psychological attribute that influences job readiness [7]. Graduates with high self-efficacy are more likely to apply for positions and engage in continuous learning, which further elevates their skill levels and career prospects [7]. Enstroem and Schmaltz (2024) [108] introduce a 'work-readiness hierarchy' that differentiates between transversal skills and employability outcomes, highlighting the significance of psychological mediators, such as self-efficacy, in linking skill acquisition to employment success.

In the realm of IT undergraduates pursuing job positions, self-efficacy pertains to their belief in their capability to successfully utilize their UI/UX skills, technical expertise, and interpersonal abilities to obtain employment in this area [106]. The conviction in one's abilities can greatly affect motivation, effort, and perseverance during the job search, ultimately shaping employability outcomes and the conceptual foundation for self-efficacy as a mediator is rooted in its impact on a person's behaviors and decisions [101,109]. IT undergraduates who possess greater confidence in their UI/UX, technical, or interpersonal skills tend to establish more ambitious career objectives, proactively pursue pertinent job opportunities, and demonstrate resilience when confronted with challenges and rejections throughout their job search [109,110]. Enstroem and Benson (2024) [111] demonstrate that enterprise education interventions foster authentic self-efficacy centered on confidence in unfamiliar tasks, rather than merely correlating with skill acquisition, highlighting the necessity of assessing self-efficacy as a separate construct indicative of task-specific confidence.

Buker (2024) [110] findings indicate that confidence in UI/UX abilities serves as a crucial mediator in the connection between having these skills and attaining UI/UX employability among IT undergraduates. This suggests that possessing robust UI/UX skills is crucial, but an individual's confidence in their capacity to apply these skills effectively in a professional environment significantly boosts their likelihood of securing employment. The incorporation of self-efficacy as a mediator in presented model aligns with prior studies indicating that self-efficacy, especially in creative or task-specific areas, acts as a conduct between competencies and innovative work behavior in knowledge-intensive sectors [112]. Consequently, it is deemed rational to presume that students' attitudes and motivations concerning employability will be affected by their self-evaluation of skills and competencies [113,114]. Students can reinforce the connection between theoretical concepts and tangible, real-world applications by acquiring practical experience through workplace engagements

during their undergraduate education. This enhances their confidence in undertaking internship assignments and elevates their overall preparedness for future professional pursuits [115].

IT undergraduates who possess strong confidence in their abilities tend to proactively pursue opportunities to utilize their skills, including engaging in design competitions, contributing to open-source projects, or taking on freelance UI/UX assignments [53,109,110,116]. These experiences enhance their skills and serve as concrete proof of their capabilities for their portfolios. Furthermore, those who possess robust self-belief in their UI/UX abilities often demonstrate greater confidence in interviews and design challenges, successfully highlighting their skills and design thinking methodologies [53,110].

Consequently, considering the evidence presented, the study posits the following hypothesis.

$H_8$ – Self efficacy has a mediating impact on the relationship between UI/UX skills and employability of IT undergraduates.

In a similar vein, confidence in one's soft skills serves as a vital intermediary in the relationship between having these skills and attaining employability for IT undergraduates [109,117,118]. The assurance in one's ability to communicate, collaborate, solve problems, and other pertinent interpersonal skills can greatly impact the effectiveness with which these skills are shown and applied in the quest for job opportunities [106,107].

IT undergraduates who possess strong self-efficacy in their soft skills tend to participate more confidently in team-based interview tasks, clearly articulate their design rationale during presentations, and actively engage in networking activities [102,107,119]. Moreover, those who possess a high level of confidence in their soft skills are more adept at managing the interpersonal elements of the job search, including conducting interviews and establishing connections with hiring managers [107,117].

Consequently, considering the evidence presented, the study posits the following hypothesis.

$H_9$ – Self efficacy has a mediating impact on the relationship between soft skills and employability of IT undergraduates.

Studies show that confidence in technical abilities plays a crucial role in linking the possession of these skills to attaining employability for IT undergraduates [7]. Confidence in coding skills, expertise with design tools, and comprehension of pertinent technologies greatly impacts the likelihood of obtaining positions that demand these technical abilities [7–9,107].

IT undergraduates who possess strong technical skills and confidence in their abilities are more inclined to engage in demanding technical projects that enrich their portfolios and showcase their competencies to prospective employers [9,98]. Furthermore, students' confidence in their technical skills motivates them to pursue positions that demand a solid technical background, thereby broadening their job prospects [7]. Benson and Enstroem's (2017) [120] findings that professional skill development and self-confidence co-evolve within undergraduate curricula, facilitating the integration of related constructs such as technical skills and self-efficacy, while acknowledging their dynamic interaction.

Consequently, considering the evidence presented, the study posits the following hypothesis.

$H_{10}$ – Self efficacy has a mediating impact on the relationship between technical skills and employability of IT undergraduates.

### Conceptual framework

Using the conceptual framework depicted in Fig 1, the purpose of this study is to determine whether the hypotheses that have been provided have an impact on the employability of undergraduate students in the field of user interface and user experience in Sri Lanka.

## 3. Methodology

This section details the research design, sampling strategy, data collection procedures, and analytical methods, structured to ensure methodological rigor and alignment with the study's objectives.

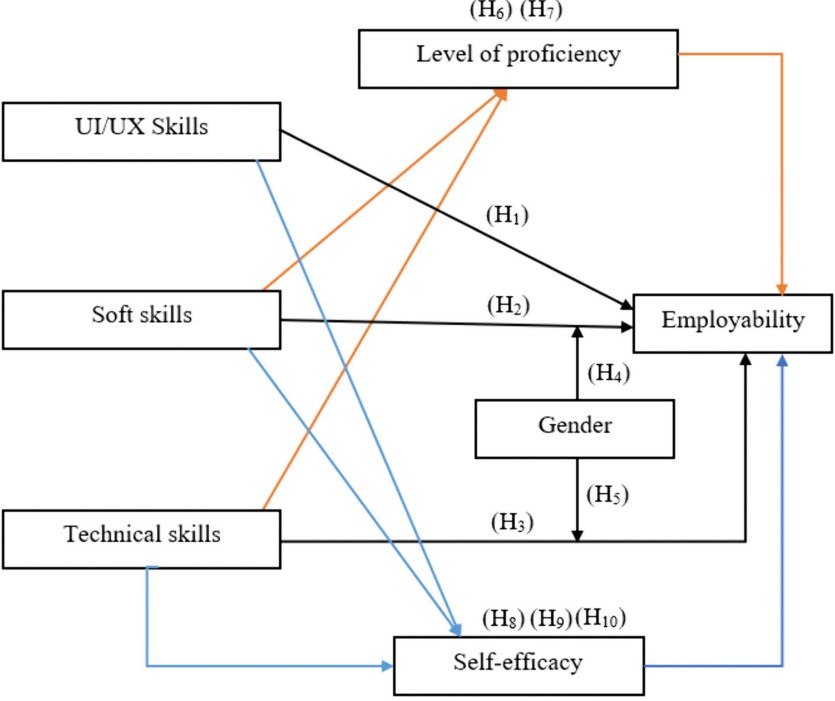

**Fig 1. Conceptual framework.**

## Measurements of variables

This section provides a comprehensive overview of the measurement methodologies utilized for the principal variables in the study: UI/UX, TS, and SS. The choice of suitable measuring scales is essential for guaranteeing validity and reliability in research findings, especially when investigating multidimensional variables.

Conceptualizing and measuring perceived employability varies widely across studies. This study assesses perceived employability by assessing self-confidence, perception of external job opportunities, field reputation, and institutional prestige. The dimensions are operationalized using validated scales [44]. Self-belief measures personal employability, while external labor market awareness measures job availability and quality. This multidimensional approach uses Likert-scale items to assess perceived employability by measuring cognitive and affective perceptions. According to Tomlinson (2008) [121], students increasingly view academic qualifications as necessary but insufficient, emphasizing the need to add skills and experiences to compete. This study measures perceived employability using self-confidence in skills, labor market awareness, and academic and institutional reputation. Recent research emphasizes the importance of digital skills in employability perceptions. Tee et al. (2024) [122] list "information and data literacy," "problem-solving," and "digital content creation" as top demands, with "communication and collaboration" emerging as a major skill gap that affects employability. These findings support assessing technical and interpersonal abilities in employability frameworks. These perspectives inform a solid, comprehensive measurement framework that captures cognitive, skill-based, and contextual aspects of perceived employability, making it relevant to current labor market realities and the IT sector's dynamic skill demands. A broadly acknowledged standardized scale for assessing professional knowledge in design-related domains does not exist, as this construct differs considerably across diverse professional settings and specialties. This study employs a multidimensional method to assess professional knowledge by integrating five essential components: design research,

design process, design principle, design quality, and information hierarchy. This measuring methodology is guided by known research frameworks defined by Borriraklert (2021) [39], and supporting literature [6,40,42,73]. The criteria for assessing UIUX abilities were chosen for their connection to modern design methodologies and their capacity to thoroughly include the extensive knowledge necessary in professional environments. Each component is evaluated using many tasks that examine both theoretical comprehension and practical application skills.

Technical skills reflect the practical, tool-oriented competences that professionals must acquire to do their duties efficiently. The assessment methodology for technical competencies includes five essential dimensions: robust portfolio development, proficiency in HTML, CSS, and JavaScript, knowledge with wireframing tools, experience with prototype tools, and expertise in user research methodologies. The present study utilizes measuring scales verified by Potter (2020) [74] and other researchers which offer extensive frameworks for evaluating technical abilities in design and development settings [60]. The measuring items utilize a blend of self-assessed competence ratings and scenario-based evaluations to capture both perceived and actual technical competencies. The evaluation of wireframing tool proficiency encompasses criteria that gauge both knowledge with particular tools and the capacity to utilize them in addressing design difficulties. García-García (2024) [123] assert that evaluation of technical skill must be consistently revised to align with advancing industry norms and technological innovations.

Soft skills denote interpersonal and self-management talents that profoundly influence professional efficacy beyond technical skills. Significant differences exist in the conceptualization and measurement of soft skills across various research. The presented methodology assesses soft skills across eight dimensions, communication, work ethic, documentation, time management, collaboration, problem-solving, decision-making, and project management utilizing measuring scales derived from previously validated instruments [65,123–125]. These scholars have created and verified extensive evaluation frameworks designed to evaluate soft skills in professional settings. The communication factor evaluates both written and vocal communication skills in many professional contexts, whereas the work ethics component examines dedication, honesty, and professional behavior.

### Research design

The study adopts a positivism research philosophy, emphasizing empirical observation, objective measurement, and statistical validation of hypotheses. Positivism aligns with the quantitative nature of this research, as it seeks to identify causal relationships between variables through systematic data analysis. A deductive approach guides the study, deriving ten hypotheses from established theories on employability capital, human capital, and social cognitive theory [126–130]. By merging these three concepts, the research creates a solid conceptual framework for analyzing the impact of user experience abilities on graduate employability. Employability capital theory presents a multidimensional and process-oriented framework, human capital theory offers the economic justification for competency enhancement, and social cognitive theory discusses the psychological and social factors that facilitate learning and career achievement. Collectively, these viewpoints provide a thorough examination of the determinants influencing employability in the digital age and guide actionable initiatives for stakeholders in higher education and business [126,130]. Furthermore, the model's theoretical framework is grounded in work readiness and social cognitive theories, emphasizing the essential function of psychological mediators in transforming skill acquisition into employability results, especially in innovation. Nevertheless, these skills are effectively utilized to enhance employability through psychological principles such as self-efficacy. Self-efficacy quantifies an individual's confidence in their ability to apply their skills in novel and challenging circumstances, which is vital in swiftly evolving technological landscapes where adaptability and innovation are imperative [131].

These hypotheses are empirically tested using structured instruments to validate or refute theoretical propositions. A quantitative methodology is adopted to operationalize latent constructs into measurable variables. This approach enables hypothesis testing through statistical techniques, ensuring generalizability and replicability.

## Population and sampling frame

The target population comprises Computer Science/IT undergraduates from Sri Lankan state and non-state universities. Using 2022/2023 admission data (N = 2,797) from the National Human Resources Development Council, the population was stratified into two standardized subgroups: state (n = 1,823) and non-state (n = 974) university students. A sample of 345 participants participated using the Krejcie and Morgan (1970) table, ensuring a 95% confidence level and ±5% margin of error. The participants were randomly selected from university registries and invited via messages to complete an online questionnaire. To mitigate non-response bias, reminders were sent at two-week intervals over a six-week period.

## Data collection and analysis

Data was collected via a structured questionnaire featuring closed-ended questions and a 5-point Likert scale from 15 November 2024–10 January 2025. The standardized format ensured consistency, facilitating efficient analysis of responses related to employability, proficiency, self-efficacy, and demographic factors.

Structural Equation Modelling (SEM) was selected for its capacity to evaluate complex relationships among multiple variables, including latent constructs and mediating/moderating effects. SEM's reflective measurement model assessed how observed survey responses represent underlying construct [130]. The analysis tested hypotheses simultaneously, accounting for direct, indirect, and moderated effects, while addressing measurement error and multiple linearity. The analysis tested hypotheses simultaneously, accounting for direct, indirect, and moderated effects, while addressing measurement error and multiple linearity.

This methodological framework balances consistency and feasibility, aligning with the study's quantitative objectives and ensuring replicability.

## Ethical consideration

Authorization to conduct the research and gather data was secured from the Institution at the beginning of the research. To ensure ethical consent, participants were provided with detailed information about the study's objectives and the questionnaire. All concerns were clarified prior to obtaining consent for participation. Explicit written consent was obtained from each respondent by ticking the consent statement in questionnaire. Participation was entirely voluntary, and only those who explicitly consented were provided with access to the questionnaire. This consent process was witnessed by the authors. The study was conducted with the ethical approval of the XXXX Institution's Ethics Review Committee.

## 4. Results

The findings provide a summary of participants identifying the demographic characteristics of the respondents. A total of 345 undergraduates engaged in the survey, with 56.2 percent identifying as male (n = 194) and 43.8 percent as female (n = 151). Their employment status indicated that 61.2 percent were employed (n = 211), 29.3 percent were unemployed (n = 101), and 9.6 percent classified as self-employed (n = 33). Regarding the kind of academic setting, 34 percent of respondents were affiliated with government institutions (n = 117), whilst 66 percent were associated with non-government universities (n = 228). The data file is provided in S1 Appendix.

## Measurement model assessment

The strength of quantitative research relies on the reliability of its measurement tools. This analysis adheres to established practices in the field by modelling all constructs UI/UX skills, technical skills, soft skills, proficiency, self-efficacy, and employability as reflective [132].

The assessment of the measurement model's reliability and validity was conducted before evaluating the structural model, applying Cronbach's Alpha (CA), Composite Reliability (CR), and Average Variance Extracted (AVE). The results validate the robustness of the measurement approach. The results are detailed in Table 1 and Fig 2 show the correlation and the PLS-SEM algorithm resulted model

In accordance with the established guidelines put forth by Hair (2021) [132], the acceptable threshold for outer loadings is determined to be 0.5, while a preferable value should surpass 0.7. The outer loadings reported varied between 0.634 and 0.942, exceeding the minimum threshold of 0.5 and clearly indicating satisfactory item-level reliability. This shows that the elements within each construct reliably assess the same fundamental idea, enhancing trust in the validity of the following analyses.

Hair (2021) [132] suggest that an AVE of at least 0.50 is deemed acceptable, signifying that the construct accounts for a minimum of half the variance in its indicators. The AVE values observed in the study varied between 0.569 and 0.871 for all constructs, thereby confirming convergent validity. This strong demonstration of convergent validity reinforces the idea that the indicators successfully reflect the underlying constructs they aim to measure, thereby enhancing the credibility of the measurement model.

## Discriminant validity

The present study verified discriminant validity through the application of the Heterotrait-Monotrait (HTMT) ratio and an analysis of cross-loadings. According to the guidelines established by Hair (2021) [132], an HTMT ratio threshold of less than 0.90 indicates sufficient discriminant validity.

The HTMT matrix indicates that the majority of values are below the 0.85 threshold, which supports the presence of adequate discriminant validity. The relationship between TS and SE demonstrated moderate correlations with other constructs, nearing but not exceeding the established threshold. Referring to Table 2, EP exhibited significant discriminant validity in relation to Gender (HTMT = 0.050) and UIUX (HTMT = 0.751). The correlation between EP and SE (HTMT = 0.967) surpassed the threshold, indicating a possible conceptual overlap between these constructs. Second, the TS and SE metrics (HTMT = 0.930) also approached the upper limit, necessitating a closer examination of their distinctiveness within the model. The interaction terms Gender × TS (HTMT = 0.316–0.809) and Gender × SS (HTMT = 0.221–0.809) demonstrated consistently low-to-moderate correlations, thereby affirming their discriminant validity from other constructs. The findings suggest that the influence of Gender on TS and SS is statistically separate from the primary constructs in the model.

## Structural model assessment

The structural model illustrates the hypothesized connections among the constructs, serving as the foundation of the study [132]. The current study evaluates the structural model that examines the connections among UIUX, TS, SS, LP, SE, Gender, and EP in the context of Sri Lankan IT undergraduates.

The assessment results involved an examination of Variance Inflation Factor (VIF) values to identify potential collinearity issues [132]. The majority of VIF values were below the acceptable threshold of 5 as presented in Table 3, suggesting that there is no collinearity issues present in the structural model.

Table 4 presents the path coefficients, t-statistics, & p-values, for hypothesis testing for independent variables.

The results presented in Table 4 demonstrate that all primary hypotheses concerning EP achieved statistical significance (p < 0.05). In the analysis of indicators, TS emerged as the most significant factor influencing EP (β = 0.893, p = 0.000), closely followed by SS (β = 0.771, p = 0.000) and UIUX (β = 0.719, p = 0.000). The skills in UIUX exhibited a notably positive total effect on EP, underscoring its significance as a crucial factor.

Table 5 presents the path coefficients, t-statistics, & p-values, for hypothesis testing for mediating variables and a moderate variable.

**Table 1. Measurement model assessment.**

| Variable | Indicator | Item code | Outer loading | CR | CA | AVE |
|---|---|---|---|---|---|---|
| Employability | | | | **0.944** | **0.943** | **0.778** |
| UI/UX skills | UIE1 | 0.826 | | | | |
| | UIE2 | 0.927 | | | | |
| Soft skills | SE1 | 0.895 | | | | |
| | SE2 | 0.865 | | | | |
| Technical skills | TE1 | 0.888 | | | | |
| | TE2 | 0.887 | | | | |
| UI/UX skills | | | | **0.963** | **0.962** | **0.726** |
| | Design process | UDP1 | 0.833 | | | |
| | | UDP2 | 0.869 | | | |
| | Design research | UDR1 | 0.761 | | | |
| | | UDR2 | 0.827 | | | |
| | Design principle | UDI1 | 0.868 | | | |
| | | UDI2 | 0.921 | | | |
| | | UDI3 | 0.848 | | | |
| | Information art | UIA1 | 0.872 | | | |
| | | UIA2 | 0.808 | | | |
| | Aesthetic value | UAV1 | 0.891 | | | |
| | | UAV2 | 0.865 | | | |
| Soft skills | | | | **0.974** | **0.973** | **0.687** |
| | Communication | SC1 | 0.736 | | | |
| | | SC2 | 0.753 | | | |
| | Collaborative working | SCW1 | 0.796 | | | |
| | | SCW2 | 0.813 | | | |
| | Problem solving | SPS1 | 0.845 | | | |
| | | SPS2 | 0.831 | | | |
| | Work ethic | SWE1 | 0.820 | | | |
| | | SWE2 | 0.868 | | | |
| | | SWE3 | 0.831 | | | |
| | Time management | STM1 | 0.895 | | | |
| | | STM2 | 0.822 | | | |
| | | STM3 | 0.814 | | | |
| | Documentation | SD1 | 0.839 | | | |
| | | SD2 | 0.771 | | | |
| | Decision making | SDM1 | 0.892 | | | |
| | | SDM2 | 0.866 | | | |
| | Project management | SPM1 | 0.864 | | | |
| | | SPM2 | 0.841 | | | |

*(Continued)*

| Variable | Indicator | Item code | Outer loading | CR | CA | AVE |
|---|---|---|---|---|---|---|
| Technical skills | | | | **0.960** | **0.956** | **0.678** |
| | Strong portfolio building | TPO1 | 0.735 | | | |
| | | TPO2 | 0.757 | | | |
| | Familiarity with HTML | TH1 | 0.752 | | | |
| | | TH2 | 0.856 | | | |
| | Wireframing tools | TW1 | 0.875 | | | |
| | | TW2 | 0.855 | | | |
| | | TW3 | 0.888 | | | |
| | Prototyping tools | TP1 | 0.877 | | | |
| | | TP2 | 0.881 | | | |
| | | TP3 | 0.812 | | | |
| | User research techniques | TUT1 | 0.803 | | | |
| | | TUT2 | 0.766 | | | |
| Level of proficiency | | | | **0.944** | **0.930** | **0.529** |
| | Soft skills | S1 | 0.751 | | | |
| | | S2 | 0.707 | | | |
| | | S3 | 0.539 | | | |
| | | S4 | 0.661 | | | |
| | | S5 | 0.714 | | | |
| | | S6 | 0.705 | | | |
| | | S7 | 0.764 | | | |
| | | S8 | 0.757 | | | |
| | Technical skills | T1 | 0.684 | | | |
| | | T2 | 0.762 | | | |
| | | T3 | 0.812 | | | |
| | | T4 | 0.788 | | | |
| | | T5 | 0.768 | | | |
| Self-efficacy | | | | **0.885** | **0.884** | **0.812** |
| | | SSE1 | 0.884 | | | |
| | | USE1 | 0.913 | | | |
| | | TSE1 | 0.907 | | | |

## 5. Discussion

The present study provides important insights into the factors that collectively influence employment outcomes for undergraduates in the information technology sector in Sri Lanka. The mechanisms encompass capabilities in UI/UX, technical expertise, interpersonal skills, proficiency, self-efficacy, and the influence of gender dynamics. This establishes a robust structural framework that combines theoretical concepts with empirical evidence through the application of PLS-SEM [132]. This framework offers a combination of methodological precision and practical significance to the discourse surrounding information technology education and employability. The following section analyses these findings in relation to existing studies, discusses the implications of these results, and offers suggestions for potential applications.

The findings indicate that skill in UI/UX (H$_1$) plays a crucial role in enhancing the employability prospects of IT undergraduates in Sri Lanka. Employers are placing greater emphasis on candidates who excel in user-centered design principles, recognizing that these abilities are essential for crafting smooth digital experiences that contribute to business

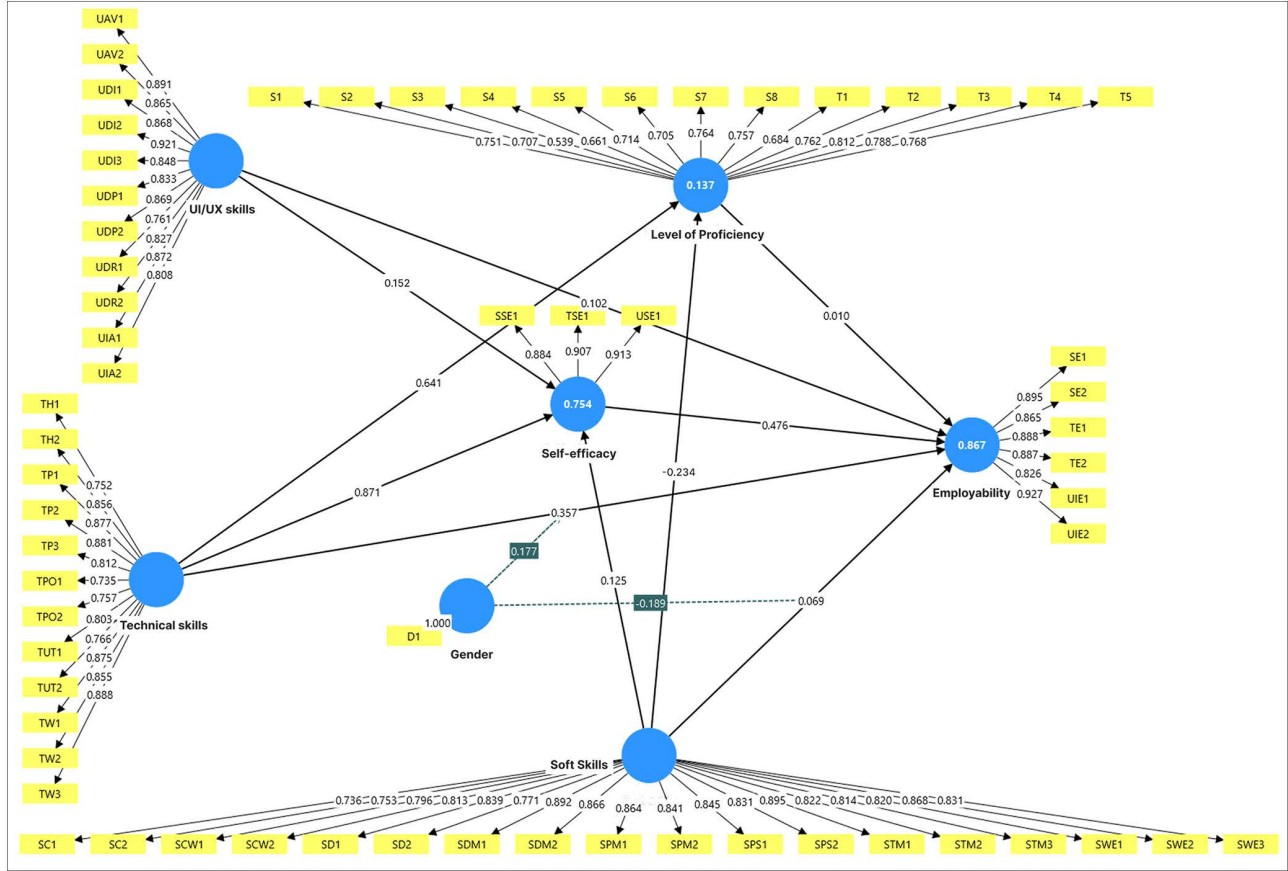

**Fig 2. Measurement model assessment.**

**Table 2. HTMT matrix results.**

|  | EP | Gender | LP | SE | SS | TS | UIUX | G x TS |
|---|---|---|---|---|---|---|---|---|
| Gender | 0.050 | | | | | | | |
| LP | 0.207 | 0.255 | | | | | | |
| SE | 0.967 | 0.052 | 0.199 | | | | | |
| SS | 0.800 | 0.142 | 0.146 | 0.788 | | | | |
| TS | 0.937 | 0.061 | 0.246 | 0.930 | 0.849 | | | |
| UIUX | 0.751 | 0.068 | 0.167 | 0.664 | 0.740 | 0.804 | | |
| Gender x TS | 0.338 | 0.008 | 0.230 | 0.327 | 0.366 | 0.316 | 0.214 | |
| Gender x SS | 0.337 | 0.038 | 0.243 | 0.411 | 0.446 | 0.376 | 0.221 | 0.809 |

Although most constructs satisfied the criteria for discriminant validity, the elevated HTMT values among EP, SE, and TS underscore the necessity for theoretical refinement to elucidate their distinct contributions in forthcoming research.

success. This outcome corresponds with previous studies, which underscore the increasing significance of UI/UX skills in the technology industry for enhancing customer happiness and competitive edge. Both Nugraha & Fatwanto (2021) [6] and Hartson (2019) [1] emphasize that UI/UX knowledge is now a fundamental need for IT positions rather than only a

**Table 3. Variance Inflation Factor Table.**

| | VIF |
|---|---|
| UIUX →EP | 2.894 |
| SS→EP | 4.110 |
| TS→EP | 7.650 |
| Gender →SS→EP | 3.377 |
| Gender →TS→EP | 2.933 |
| SS→LP→EP | 3.056 |
| TS→LP→EP | 3.056 |
| UIUX→SE→EP | 2.600 |
| SS→SE→EP | 3.229 |
| TS→SE→EP | 3.867 |

**Table 4. Structural model assessment (Independent variables).**

| | Path | Sample mean | SD | t value | p value | CI | | Decision |
|---|---|---|---|---|---|---|---|---|
| | | | | | | 2.5% | 98% | |
| H$_1$ | UIUX →EP | 0.716 | 0.040 | 18.192 | 0.001 | 0.041 | 0.166 | Supported |
| H$_2$ | SS→EP | 0.768 | 0.042 | 18.552 | 0.000 | 0.041 | 0.194 | Supported |
| H$_3$ | TS→EP | 0.891 | 0.018 | 50.656 | 0.000 | 0.242 | 0.487 | Supported |

**Table 5. Structural model assessment (Mediating and moderate variables).**

| | Path | Sample mean | SD | t value | p value | CI | | Decision |
|---|---|---|---|---|---|---|---|---|
| | | | | | | 2.50% | 98% | |
| H$_4$ | Gender →SS→EP | 0.317 | 0.089 | 3.667 | 0.000 | 0.289 | 0.087 | Supported |
| H$_5$ | Gender →TS→EP | 0.317 | 0.088 | 3.706 | 0.000 | 0.098 | 0.256 | Supported |
| H$_6$ | SS→LP→EP | 0.062 | 0.093 | 0.59 | 0.555 | −0.713 | 0.241 | Not supported |
| H$_7$ | TS→LP→EP | 0.267 | 0.087 | 2.932 | 0.003 | 0.554 | 0.759 | Supported |
| H$_8$ | UIUX→SE→EP | 0.611 | 0.051 | 12.038 | 0.003 | 0.058 | 0.25 | Supported |
| H$_9$ | SS→SE→EP | 0.733 | 0.043 | 17.132 | 0.000 | 0.008 | 0.254 | Supported |
| H$_{10}$ | TS→SE→EP | 0.861 | 0.021 | 40.627 | 0.000 | 0.738 | 1.006 | Supported |

The interaction effects revealed noteworthy relationships, as Gender moderated the influence of SS ($\beta = 0.327$, $p = 0.000$) and TS ($\beta = 0.328$, $p = 0.000$) on EP. In terms of SE, the most significant factor was TS ($\beta = 0.862$, $p = 0.000$), followed by SS ($\beta = 0.736$, $p = 0.000$) and UIUX ($\beta = 0.616$, $p = 0.000$). The analysis revealed that TS had a notable impact on LP ($\beta = 0.255$, $p = 0.003$), while SS did not demonstrate a significant effect ($\beta = 0.055$, $p = 0.555$) on LP.

benefit. Past research support finding that the fundamental reason is that the digital revolution of enterprises has heightened the demand for intuitive, accessible, and engaging digital interfaces [40,42,52,73]. Organizations acknowledge that exceptional UI/UX significantly influences user retention and product success, rendering these talents highly desirable in recruiting processes [1,37,58].

The study provided robust validation for the hypothesis concerning soft skills (H$_2$). Key determinants of employability include communication, adaptability, teamwork, and critical thinking [2]. This discovery aligns with extensive literature conducted by Patacsil & Tablatin (2017) [60] and Bisschoff (2024) [21] substantiates that the integration of soft skills with technical expertise markedly enhances the employment chances of graduates. In past research employers highlighted

that these interpersonal skills are essential for playing through complex work settings and working together within teams [2,21,60,67]. This finding aligns with studies emphasizing the importance of soft skills in combining with technical knowledge to improve employability [60,74]. The findings of Mata et al. (2021) [72] indicate that time management correlates with improved performance outcomes, thereby justifying our classification of time management as an essential soft skill. Efficient time management is essential for employability as it mitigates stress and enhances productivity. [77]. It assists individuals in organizing and executing tasks efficiently, thereby enhancing productivity and reducing stress levels [77,133]. Communication, adaptability, collaboration, and analytical reasoning are essential determinants of employability [70]. These results underscore the significance of soft skills in securing employment. Career success is significantly influenced by employability skills, particularly self-management and job performance [134].

In Sri Lanka, many graduates, however, do not receive sufficient training in soft skills through their academic programs. To tackle this gap, it is essential to incorporate the development of soft skills into IT curricula via workshops and mentorship initiatives [3,135,136]. Graduates must possess the ability to articulate ideas, adapt to change, and collaborate effectively. The absence of organized soft skills training in academic curricula intensifies this disparity, highlighting the necessity for focused initiatives like seminars and mentorships to adequately equip students for the labor market.

The findings corroborate the hypothesis that technical skills ($H_3$) play a crucial role in enhancing employability. The results align with research from other developing nations, including Vietnam and India, where targeted training initiatives have demonstrated improvements in employability results [36,137]. Research from Sri Lanka and other developing nations substantiates that focused technical training efforts have resulted in enhanced employment results [2,113]. Further research by Potter (2020) [74], Thang (2016) [137], and Getto (2023) [20], similarly emphasize the need of technical proficiency in IT positions and designing positions. The primary drive is the fast advancement of technology and the ensuing necessity for current technical proficiency in the sector. Nonetheless, Getto's (2023) [20] study revealed that technical skills alone are inadequate. Employers are progressively looking for candidates who can combine technical skills with innovative problem-solving capabilities, an essential skill set for UI/UX positions [20,40,58].

Misra and Khurana (2017) [138] underscored the necessity for advanced technical competencies within employability frameworks, defining technical skills as one of six critical skill sets requisite for IT professionals to fulfill industry requirements. Puwakgahawela et al. (2023) [139] identify significant discrepancies between employers' expectations and graduates' technical competencies, underscoring the imperative for colleges to modify their curricula to align with industry requirements. Moreover, studies demonstrate that technical skills are crucial for career progression, employment opportunities, and adaptability in rapidly changing IT landscapes. All peer-reviewed studies concur that technical skills are vital for the employability of IT undergraduates, and that continuous curriculum updates and practical training are required to align with industry standards.

The findings demonstrate that gender plays a significant role in moderating the relationship between soft skills ($H_4$), technical skills ($H_5$), and employability. This indicates that the influence of these skills on obtaining employment varies between male and female IT graduates. This discovery corroborates other studies suggesting that employers may evaluate skill sets variably according to gender, resulting in inequitable job prospects. Previous studies indicated that the employability abilities of male and female IT undergraduates considerably varied following exposure to industrial job experience [82,91,106,109]. Perera (2024) [82] and Jayasinghe (2020) [3] discover that gender stereotypes and biases substantially affect the employability of Sri Lankan undergraduates, which affect hiring practices and workplace interactions. Such biases may influence the assessment and integration of male and female graduates into teams, requiring specific policies and educational changes to foster inclusion and equal opportunity. This discovery underscores the possibility of gender-related inequalities within the IT sector in Sri Lanka

Carter (2011) [140] and Groeneveld, Becker, and Vennekens (2020) [125] assert that soft skills such as communication, teamwork, adaptability, and problem-solving are increasingly vital for the employability of IT graduates. Studies demonstrate that female students often exhibit heightened confidence in soft skills, potentially enhancing their employability by

counteracting gender biases in technical domains [83,92,141]. These skills assist recent female graduates in maneuvering through workplace dynamics and overcoming structural barriers. Conversely, soft skills may hold lesser significance for the employability of male students compared to technical skills. This gender moderation indicates that to enhance comprehensive career readiness, employability programs must emphasize the cultivation of soft skills alongside technical training, especially for female IT students. Gender, however, influences this relationship: while core technical skills do not significantly differ between male and female students, the pervasive gender biases in the tech industry frequently hinder female graduates from converting these skills into employment opportunities [83,89,90]. Targeted interventions are essential to enhance the technical self-efficacy and workplace acceptance of female IT students, as these biases can undermine women's confidence and acknowledgment in technical positions [142]. Consequently, while technical skills are crucial, gendered social and organizational factors affect their employability.

The observed lack of equality in proficiency's mediational role wherein proficiency significantly mediates the technical skills, employability relationship ($H_7$) but fails to mediate the soft skills, employability relationship ($H_6$) reveals fundamental differences in how these skill domains translate into employability outcomes within UI/UX design context. Samarasinghe (2022) [2] emphasizes that employers in the IT sector are placing greater importance on candidates who possess both technical skills and effective interpersonal abilities. In the existing research, it has been found that the competency of technical abilities has a considerable favorable influence on the employability of students studying information technology [94,98,100,106]. Proficiency, conceptualized as the demonstrated mastery and practical application of competencies beyond mere acquisition, serves as a critical bridge for technical skills [143]. The finding aligns with competency-based frameworks in IT education, where technical proficiency functions as a controller converting latent knowledge into verifiable performance metrics essential for competitive fields like UI/UX design [144,145]. The mediating role of proficiency in the technical skills and perceived employability pathway reflects the fundamentally graduated, measurable character of technical competencies. In design education, key technical indicators are all domains in which performance exists on a clear scale from novice to expert [146]. Students can therefore appraise their own employability not simply from possessing skills in tools such as Figma or prototyping platforms, but from their subjective assessment of how well they perform with them, suggesting that skill possession alone is an insufficient predictor of perceived employability, and that an intermediate evaluation of proficiency level is what does the explanatory work [24].

However, the analysis did not find evidence to support a similar mediating effect in the relationship between soft skills and employability in UI/UX ($H_6$). This indicates that while possessing a high level of technical proficiency significantly enhances the likelihood of securing a UI/UX-related position, it may not suffice for soft skills. Despite previous research indicating that competence in soft skills is considered essential by employers, survey participants assessed proactivity as the least competent soft skill for UI/UX employability. Although several studies indicate the beneficial effects of soft skill mastery, Dubey (2022) [147] reveals that IT students have a weakness in these skills, suggesting an insignificant influence on employability. The subsequent study reveals that the majority of students possess proficient documentation abilities due to their regular involvement in university projects and assignments. Among STEM undergraduates, technical proficiency mediated skill to employability perceptions, while soft skill proficiency showed direct effects absent mediation, attributed to assessment biases in self-reports [148,149]. In creative tech fields similar to UI/UX, proficiency in tools like Figma correlated with employability perceptions, but communication mastery did not, as employers prioritize demonstrated teamwork over self-rated aptitude [150–152]. The findings emphasize the necessity for IT curricula to prioritize the integration of soft skills training with practical application in real-world scenarios, as this integration could be more closely linked to employability outcomes in UI/UX. These findings carry direct implications for UI/UX curriculum design. Curricular designs should incorporate reflective support, mentor guided supervision, and task role coherence to ensure that learning environments serve as meaningful developmental space where an approach that allows technical proficiency to deepen deliberately while soft skills grow through authentic professional encounter [34].

The findings indicated that self-efficacy served as a mediator in the relationships among UI/UX skills ($H_8$), soft skills ($H_9$), technical skills ($H_{10}$), and the employability of IT undergraduates. These findings support earlier studies highlighting the essential importance of self-efficacy in converting skills into concrete career results [53,109,113,118]. Individuals who possess a strong belief in their abilities tend to proactively pursue job opportunities, effectively showcase their skills, and remain resilient when confronted with obstacles. According to Da Motta Veiga and Turban (2018) [103] and Ngoc et al. (2024) [109], self-efficacy strengthens the intensity of job search efforts and increases the likelihood of success. This is because it enables individuals to transform their talents into measurable professional results [109,110,113,118]. At previous studies, it was shown that self-efficacy acts as a mediator between the link between adoption of information and communication technology and employability among undergraduate students at universities [7–9,98,109,153]. Recent studies highlight that proficiency as the demonstrated mastery and practical application of competencies that acts as a crucial bridge between skill acquisition and employability outcomes, especially in fields such as UI/UX design where technical expertise must be coupled with effective interpersonal abilities [154]. This mediating role underscores that perceived employability hinges on the ability to convert latent knowledge into observable performance, not simply on confidence or the static inventory of skills [24].

There is more to securing a job in the field of information technology than simply possessing border skills; individuals also need to possess talent self-efficacy. The fundamental explanation behind this is that graduates who have a high level of self-efficacy are better able to successfully exploit their skills in the job market [53,101,110,116]. Because individuals who have a strong sense of self-belief are more likely to seek opportunities, persevere through failures, and adjust to changing demands, self-efficacy is an important target for educational interventions such as mentorship and experiential learning [107,109,119]. This highlights the significance of cultivating a nurturing educational atmosphere that empowers learners to build self-assurance in their skills. This highlights the significance of cultivating a nurturing educational atmosphere that empowers learners to build self-assurance in their skills.

Previous research has emphasized the co-evolution of competence and confidence in educational and occupational development [111,120]. The significance of educational interventions that concurrently enhance students' technical skills and cultivate their self-efficacy in tackling new and challenging tasks is underscored by the distinct yet complementary functions of these elements. The study by Chughtai and Khan (2024) [131] indicates that knowledge-oriented leadership influences employees' innovative performance through mediators such as work engagement and knowledge-sharing behavior, with creative self-efficacy acting as a significant moderator. Their moderated mediation model posits that psychological variables both mediate and amplify the impact of knowledge and skills on creative outcomes within the IT sector. This closely mirrors the findings, demonstrating that the relationship between skills and employability is mediated by self-efficacy, suggesting that confidence enhances the translation of skills into tangible career advantages.

The findings of Haider et al. (2023) [155], which indicate that knowledge sharing, and personal confidence are crucial mechanisms fostering innovation in project-based sectors, align closely with the results demonstrating that self-efficacy mediates the relationship between employability and soft skills among IT undergraduates. Haider and colleagues found that the influence of ambidextrous leadership on creative work behavior is mediated by knowledge sharing. This underscores the significance of self-confidence and belief in one's capabilities in employing communication and interpersonal skills to excel in dynamic, technologically advanced environments.

## 6. Implications for it education in Sri Lanka

The results of the study hold considerable importance for the field of IT education in Sri Lanka. The findings emphasize the importance of a comprehensive strategy in curriculum development that encompasses technical skills alongside soft skills, proficiency levels, and self-efficacy. Integrating training programs that are sensitive to gender, along with practical experience and interventions designed to enhance self-confidence, will enable IT institutions in Sri Lanka to more effectively equip their graduates for success in the highly competitive UI/UX job market. Future investigations ought to delve

deeper into these relationships, taking into account further moderating and mediating factors that could impact on the employability of IT graduates.

## 7. Limitations of the study

The research focuses solely on undergraduate students who are currently enrolled in degree programs in the fields of computer science and information technology in Colombo, Sri Lanka. As a result, it is possible that the findings cannot be generalized to other academic subjects, geographical regions, or even various regions within Sri Lanka. For the purpose of this research, only data from undergraduate students is collected. The viewpoints of employers, faculty members, and other stakeholders are not included, despite the fact that they may provide useful insights into the demand for user interface and user experience skills. Students are asked to self-evaluate information about their experiences and talents, and this information is used in part to inform the research. This increases the likelihood of response bias, which may include a bias toward social want or recollection bias, which may result in the data being less accurate than it otherwise would have been. The research focuses on user interface and user experience skills and how they affect employability. This study is constrained by its reliance on cross-sectional, self-reported data collected during a single session which may result in common method variance (CMV). This bias could amplify correlations and path coefficients possibly inflating the connection between technical Skills and employability. The analyses provide valuable insights into the absence of control for CMV and the challenge in establishing causality limit the interpretation of the findings. Future investigations should employ longitudinal designs or multi-source data to mitigate common method variance.

## 8. Suggestions for future research

Future research might go into several different areas, according to some proposals, to better understand the employability factors for IT graduates in Sri Lanka's UI/UX industry. First, carrying out long-term studies that track the career progression of graduates in the information technology sector over time can provide valuable insights into how their skills impact their long-term employability and job satisfaction. Work skills development is crucial for students to monitor their progress and assess their skill levels [156]. Possessing a robust set of skills is not merely advantageous but fundamentally essential for navigating the complexities and demands of the contemporary professional landscape [157].

Second, an investigation that examines the effectiveness of user interface and user experience education across different regions or countries can aid in pinpointing the best practices and innovative approaches to curriculum development that are customized to address the needs of the local industry. Future studies should examine the applicability of employability models across diverse cultural and geographic contexts. Frameworks that examine the global variations in entrepreneurial skill sets and adaptive behaviors, such as those proposed by Tehseen et al. (2024) [158], offer valuable guidance on competitive advantage. Moreover, employability is mediated variably across countries by components of social cognitive career theory, such as self-efficacy, underscoring the necessity of considering cultural nuances in the development of skills and employability interventions.

Third, investigating the effectiveness of various experiential learning methodologies, including internships, co-op programs, and project-based learning, on the preparedness of students for careers in user interface and user experience can contribute significantly to enhancing educational practices. Finally, such research endeavors will provide critical evaluations of pedagogical approaches and specific interventions, offering actionable insights for educators to optimize curricula and teaching methods to ensure IT students are well-equipped with the employability skills demanded by the industry.

## 9. Conclusion

The investigation provides essential insights into how UI/UX competencies, technical skills, and soft skills converge to influence the employability of IT graduates in Sri Lanka, highlighting important gaps in existing literature. The findings emphasize the increasing need for proficient UI/UX professionals worldwide and point out the distinct challenges

encountered by developing economies such as Sri Lanka in synchronizing educational results with industry needs. The conducted research emphasizes the viewpoints of IT undergraduates, connecting theoretical frameworks with practical realities and offering a detailed understanding of the influence of skill development on employability within the local context.

The findings indicate that securing a position in UI/UX design is influenced not only by technical skills but also significantly relies on interpersonal abilities, creativity, adaptability, and self-confidence. It highlights that although technical skills are essential for UI/UX careers, soft skills like communication, problem-solving, and teamwork is also vital for achieving professional success. Moreover, the study emphasizes the significant impact of proficiency and self-efficacy on employability, providing important insights into the ways in which confidence and skill mastery affect career preparedness. The research explores gender dynamics in the UI/UX sector, enhancing our understanding of potential inequities and laying the groundwork for targeted interventions aimed at fostering inclusivity.

In summary, the result of the study plays a crucial role in enhancing our comprehension of how IT education in Sri Lanka can more effectively equip graduates for careers in UI/UX design. By tackling deficiencies in skill development and employability frameworks, it lays the groundwork for significant transformations in educational methodologies and industry collaboration. The presented findings significantly contribute to academic discussions while also providing a practical resource for stakeholders looking to cultivate a vibrant UI/UX talent pool in Sri Lanka's digital economy.

## Supporting information

**S1 Appendix. Data file.**
(XLSX)

## Author contributions

**Conceptualization:** Dhamindi Senadheera, Krishantha Wisenthige.

**Data curation:** Dhamindi Senadheera.

**Formal analysis:** Dhamindi Senadheera.

**Investigation:** Dhamindi Senadheera.

**Methodology:** Dhamindi Senadheera, Krishantha Wisenthige.

**Project administration:** Krishantha Wisenthige.

**Supervision:** Krishantha Wisenthige.

**Validation:** Dhamindi Senadheera.

**Visualization:** Dhamindi Senadheera.

**Writing – original draft:** Dhamindi Senadheera.

**Writing – review & editing:** Dhamindi Senadheera, Krishantha Wisenthige.

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
