## [Decision Letter · Decision Letter 0]

10 Jun 2025

Dear Dr. Wisenthige,

Thank you for submitting your manuscript to PLOS ONE. After careful consideration, we feel that it has merit but does not fully meet PLOS ONE’s publication criteria as it currently stands. Therefore, we invite you to submit a revised version of the manuscript that addresses the points raised during the review process.

We look forward to receiving your revised manuscript.

Kind regards,

Hira Salahuddin Khan, Ph.D

Academic Editor

PLOS ONE

Journal Requirements:

2. Please ensure that you refer to Figure 2 in your text as, if accepted, production will need this reference to link the reader to the figure.

3. We are unable to open your Supporting Information file [Title Page.docx]. Please kindly revise as necessary and re-upload.

Reviewers' comments:

Reviewer's Responses to Questions

**Comments to the Author**

1. Is the manuscript technically sound, and do the data support the conclusions?

Reviewer #1: Yes

Reviewer #2: No

2. Has the statistical analysis been performed appropriately and rigorously?

Reviewer #1: Yes

Reviewer #2: No

3. Have the authors made all data underlying the findings in their manuscript fully available?

Reviewer #1: Yes

Reviewer #2: Yes

4. Is the manuscript presented in an intelligible fashion and written in standard English?

Reviewer #1: Yes

Reviewer #2: No

Reviewer #1: Assessing the Influence of Diverse Skills on Employability Outcomes for IT Undergraduates

This study offers a valuable contribution by examining how UI/UX, soft, and technical skills influence IT graduates’ employability in Sri Lanka. The integration of gender as a moderator and self-efficacy and proficiency as mediators within a PLS-SEM framework makes the study theoretically rich and methodologically robust. The research fills a key gap by contextualizing employability analysis in a South Asian, skill-focused, and digital economy setting.

I recommend acceptance with minor revisions, particularly in terms of reducing redundancy, improving phrasing, and enhancing the manuscript's theoretical framework by incorporating recent scholarly literature.

Minor Revisions

• Reword abstract for smoother expression and remove repetition in phrases like “examines the combined effect… instead of looking at them separately.”

• Improve flow in introduction and avoid repeating the point that “Western contexts dominate existing research.”

• Clarify timeline expressions like: “Data collection was started in 15/11/24…” → “Data were collected from 15 November 2024 to 10 January 2025.”

• Clean up discussion section by removing duplicated explanations (e.g., “gender stereotypes...” repeated twice).

Integration of Literature

1. Khan, H. S. U., Li, P., et al. (2023)

Use in your paper:

Section 2.4 – Mediating Impact of Self-Efficacy

Use this study to support your argument that creative self-efficacy plays a mediating role between skill possession and work performance outcomes.

Suggested sentence:

“Prior research confirms that self-efficacy, especially in creative or task-specific domains, acts as a bridge between competencies and innovative work behavior in knowledge-driven fields (Khan et al., 2023), aligning with our model’s inclusion of self-efficacy as a mediator.”

2. Khan, H. S. U., Ma, Z., et al. (2023)

Use in your paper:

Methodology or Section 3 – Research Design / Section 5 – Discussion

This study can be cited to justify your use of a moderated mediation SEM model and the psychological variables involved, such as self-efficacy.

Suggested sentence:

“Our structural model, which includes both mediation and moderation effects, follows the precedent of recent psychological studies using similar SEM frameworks to explore layered behavioral outcomes (Khan et al., 2023).”

3. Chughtai, M. S. & Khan, H. S. U. (2023)

Use in your paper:

Section 2 – Literature Review and Section 4 – Results/Discussion

This paper can reinforce your model’s theoretical basis and support discussion around the indirect effects of skills via psychological mediators, particularly in innovation or tech-based job roles.

Suggested sentence:

“Research by Chughtai and Khan (2023) highlights how knowledge-focused leadership and individual psychological traits such as self-efficacy interact in moderated mediation models to explain innovative performance, which closely parallels our findings on the influence of soft and technical skills on employability outcomes.”

4. Yasmin, F., et al. (2024)

Use in your paper:

Section 2.1 – Employability and Skill Sets / Section 5 – Discussion

Use this to connect employability outcomes to productivity gains and HR-driven skill development.

Suggested sentence:

“Yasmin et al. (2024) emphasize that organizational performance is significantly influenced by HRM practices that develop worker skills and productivity, paralleling our argument that employability is driven by skill readiness shaped through targeted education and training.”

5. Haider, S. A., et al. (2023)

Use in your paper:

Section 2.4 – Self-Efficacy / Section 5 – Discussion

Incorporate this to support the role of knowledge-sharing behavior as a catalyst for self-efficacy and performance outcomes.

Suggested sentence:

“Our results on self-efficacy’s mediating role resonate with Haider et al. (2023), who found that knowledge-sharing and individual confidence drive innovation in project-oriented sectors.”

6. Ali, A., et al. (2021)

Use in your paper:

Section 2.1.3 – Technical Skills and Employability

This fits well to reinforce the link between IT skill acquisition and employability in governance or digital transformation roles.

Suggested sentence:

“Ali et al. (2021) found that strong IT governance frameworks rely on innovation-ready technical personnel, underscoring the importance of technical skillsets for public and private sector employability.”

7. Sohail & Ilyas (2018)

Use in your paper:

Section 2.1 – Employability and Career Outcomes

Connect this study to downstream outcomes of employability—namely, workplace retention and engagement.

Suggested sentence:

“Employability not only influences hiring prospects but also long-term outcomes like organizational commitment (Sohail & Ilyas, 2018), highlighting the importance of foundational skill development for career stability.”

8. Sohail & Rehman (2015)

Use in your paper:

Section 2.4 – Self-Efficacy

Support the idea that self-efficacy can buffer against employment-related stress.

Suggested sentence:

“Psychological resilience, such as self-efficacy, has been shown to moderate workplace stress effects and support healthier professional integration (Sohail & Rehman, 2015).”

Tehseen et al. (2024)

Use in your paper:

Section 6 – Implications or Section 8 – Future Research

Use this when addressing the transferability of your model across regions or cultures.

Suggested sentence:

“Future cross-national studies may benefit from frameworks like Tehseen et al. (2024), who examined how entrepreneurial skillsets vary across countries and influence competitive advantage.”

10. Mata et al. (2021)

Use in your paper:

Section 2.1.2 – Soft Skills / Section 5 – Discussion

This supports inclusion of time management as a key soft skill linked to performance outcomes.

Suggested sentence:

“Mata et al. (2021) demonstrate that effective time management significantly enhances individual performance under workplace stress, reinforcing our inclusion of time as a core soft skill for employability.”

Final Recommendation

The manuscript is a strong contribution to the field of digital workforce employability in developing economies. With the above textual refinements and integration of supporting literature, it will be well-positioned for publication.

Reviewer #2: This is an interesting and relevant paper, which ties into the literature on work readiness. However, the manuscript requires a substantial revision before it is publishable. Below, I have outlined some of the main concerns, with suggestions to strengthen the paper:

Your structural model includes three antecedent skill constructs (UI/UX, soft, technical), two mediators (self-efficacy and proficiency), and gender interactions. Several latent variables (e.g., Soft Skills vs Soft-Skill Proficiency, Technical Skills vs Technical-Skill Proficiency, and Skills vs Self-Efficacy) overlap conceptually and empirically, leading to very high intercorrelations (HTMT > 0.90) and collinearity (VIF > 7).

For instance, measuring ‘I can communicate effectively’ (soft skill) alongside ‘I am highly proficient in communication’ (proficiency) will produce almost identical responses, which means that the mediator-test (soft skills → proficiency → employability) becomes meaningless.

Simplify by merging redundant constructs. For instance, combine Soft Skills and their corresponding proficiency items into a single Soft-Skill Competence factor. Likewise, merge Technical Skills with Technical-Skill Proficiency, and ensure Self-Efficacy items capture confidence in performing tasks beyond current ability. Reducing the number of latent variables will alleviate identification issues and produce more stable path estimates.

Several constructs are treated as unidimensional despite comprising heterogeneous subdomains (e.g., eight soft-skill dimensions combined into one factor; five UI/UX subdomains collapsed without hierarchical testing). Some subscales have only two indicators, which raises questions about identification and reliability.

Conduct a standalone factor analysis to establish that each latent variable has at least three coherent indicators or is a justified two-indicator factor. For broad constructs (e.g., Soft Skills), either retain only the most relevant subdomains (e.g., communication, teamwork, problem-solving) or adopt a second-order factor structure: first demonstrate that each subdimension is unidimensional, then confirm that they load onto a higher-order Soft-Skill Competence factor.

All variables are self-reported in one sitting. As a result, method variance likely inflates intercorrelations. Moreover, causal language (‘Skill A influences Employability through Self-Efficacy’) is used despite cross-sectional data. The very large path coefficient from Technical Skills to Employability ( ≈ 0.89) may partly reflect common-method bias rather than a true causal effect.

In the revised manuscript, explicitly acknowledge CMV as a limitation; perform a post hoc diagnostic (e.g., Harman’s one-factor test or inclusion of a common latent factor in PLS-SEM) and report the proportion of variance attributable to method effects; and rephrase all causal claims as associations (e.g., ‘Technical skills are associated with perceived employability’). If possible, describe in the Methods how future data collection could separate measures temporally or methodologically to reduce CMV.

The literature review summarizes existing studies without critically comparing theoretical frameworks or engaging deeply with regional research. Hypotheses are stated based on generic claims rather than being grounded in specific empirical findings.

Reorganize the review into clear themes:

Definitions and measures of employability (e.g., distinguish between perceived employability and actual placement outcomes).

Empirical evidence on UI/UX, soft, and technical skills, drawing on global and South Asian studies.

Mediators (self-efficacy, proficiency) and moderators (gender, institutional type), explicitly linking each hypothesis to at least two prior studies.

In particular, please consider citing Enstroem & Schmaltz (2024) (https://doi.org/10.1108/JWAM-10-2023-0100), which offers a ‘work-readiness hierarchy’ that distinguishes transversal skills from employability outcomes, and Enstroem & Benson (2024) (https://doi.org/10.1108/ET-01-2022-0009), which demonstrates how enterprise education interventions build genuine self-efficacy rather than overlapping with skill possession. Discuss how these frameworks and findings could inform your measurement of self-efficacy (e.g., focusing on confidence in novel tasks) and your practical recommendations for curriculum design. Also, consider citing Benson & Enstroem (2017) (https://doi.org/10.1108/JMD-08-2016-0148), which shows that undergraduate students’ professional‐skill development and self-confidence are empirically intertwined. Their findings support merging overlapping constructs (e.g., technical skills + self-efficacy) and illustrate how competence and confidence co-evolve in curriculum settings

The manuscript’s current laundry list of relationships (three skill types, two mediators, two moderators) creates a disjointed narrative, and the practical implications are very general.

After merging overlapping constructs and ensuring discriminant validity, focus on the two or three strongest antecedents that remain distinct. For example, if multicollinearity prevents distinguishing Technical Skills from Self-Efficacy, consider reporting a combined Technical-Self-Efficacy factor. Present a simplified PLS-SEM with clearer, theory-driven paths (e.g., UI/UX skills → Self-Efficacy → Employability, Soft-Skill Competence → Employability) and, if you wish to explore moderation, run multi-group analyses (e.g., by gender). Finally, revise the Discussion to offer targeted recommendations, such as integrating enterprise-education modules (Enstroem & Benson, 2024) to build self-efficacy through hands-on projects, or leveraging adaptive learning platforms (Enstroem & Schmaltz, 2024) to scale soft-skill development in large classes.

Conclusion

Your manuscript addresses an important gap: examining multiple skill domains in a South Asian context. However, it requires substantial revision to achieve conceptual coherence, measurement validity, and a clear theoretical narrative.

**Do you want your identity to be public for this peer review?** For information about this choice, including consent withdrawal, please see our For information about this choice, including consent withdrawal, please see our Privacy Policy .

Reviewer #1: No

Reviewer #2: No

While revising your submission, please upload your figure files to the Preflight Analysis and Conversion Engine (PACE) digital diagnostic tool, https://pacev2.apexcovantage.com/ . PACE helps ensure that figures meet PLOS requirements. To use PACE, you must first register as a user. Registration is free. Then, login and navigate to the UPLOAD tab, where you will find detailed instructions on how to use the tool. If you encounter any issues or have any questions when using PACE, please email PLOS at . PACE helps ensure that figures meet PLOS requirements. To use PACE, you must first register as a user. Registration is free. Then, login and navigate to the UPLOAD tab, where you will find detailed instructions on how to use the tool. If you encounter any issues or have any questions when using PACE, please email PLOS at figures@plos.org . Please note that Supporting Information files do not need this step.. Please note that Supporting Information files do not need this step.

---

## [Author Response · Author response to Decision Letter 1]

18 Jul 2025

Authors have addressed comments and suggestions by reviewers and editors and provided them in a separate file (Responses to reviewer document and attached).

---

## [Decision Letter · Decision Letter 1]

10 Feb 2026

Assessing the Influence of Diverse Skills on Employability Outcomes for IT Undergraduates

PLOS One

Dear Dr. Wisenthige,

Thank you for submitting your manuscript to PLOS ONE. After careful consideration, we feel that it has merit but does not fully meet PLOS ONE’s publication criteria as it currently stands. Therefore, we invite you to submit a revised version of the manuscript that addresses the points raised during the review process.

https://journals.plos.org/plosone/s/submission-guidelines#loc-laboratory-protocols . Additionally, PLOS ONE offers an option for publishing peer-reviewed Lab Protocol articles, which describe protocols hosted on protocols.io. Read more information on sharing protocols at . Additionally, PLOS ONE offers an option for publishing peer-reviewed Lab Protocol articles, which describe protocols hosted on protocols.io. Read more information on sharing protocols at https://plos.org/protocols?utm_medium=editorial-email&utm_source=authorletters&utm_campaign=protocols ..

We look forward to receiving your revised manuscript.

Kind regards,

Rea Lavi

Academic Editor

PLOS One

Journal Requirements:

Additional Editor Comments:

Dear Authors,

Thank you for submitting your revised manuscript.

Please attend to Reviewer #3's comments. Mainly:

- Strengthen your discussion of findings based on the reviewer #3's comments.

- Fix grammar, punctuation errors, and repeated/redundant phrases where relevant.

Also, the text in the figures you supplied in the appendix is difficult to read. Check if you can provide more high-resolution images.

Best wishes

Reviewer's Responses to Questions

**Comments to the Author**

Reviewer #2: All comments have been addressed

Reviewer #3: All comments have been addressed

2. Is the manuscript technically sound, and do the data support the conclusions?

Reviewer #2: Yes

Reviewer #3: Yes

3. Has the statistical analysis been performed appropriately and rigorously?

Reviewer #2: Yes

Reviewer #3: Yes

4. Have the authors made all data underlying the findings in their manuscript fully available?

Reviewer #2: Yes

Reviewer #3: Yes

5. Is the manuscript presented in an intelligible fashion and written in standard English?

Reviewer #2: Yes

Reviewer #3: Yes

Reviewer #2: The authors have responded appropriately to the earlier comments, and the revisions have strengthened the manuscript. It provides a valuable contribution to the discourse on graduate skill sets.

I recommend it for publication.

Reviewer #3: This manuscript examines the combined influence of UI/UX skills, technical skills, and soft skills on the employability of IT undergraduates in Sri Lanka, incorporating self-efficacy and proficiency as mediators and gender as a moderator. The topic is relevant, and the study aligns well with PLOS ONE’s scope by addressing an under-researched regional context using a transparent quantitative approach.

From a methodological perspective, the study is technically sound. The sample size (n = 345) is appropriate, the use of PLS-SEM is justified given the model complexity, and the authors report reliability, convergent validity, discriminant validity, and collinearity diagnostics in line with established guidelines. The hypotheses are clearly stated, and the results generally support the conclusions drawn.

That said, I offer the following points for improvement, which I believe would strengthen clarity and robustness without requiring major additional analyses:

Conceptual overlap between employability, self-efficacy, and technical skills

The HTMT values indicate high correlations between employability, self-efficacy, and technical skills, with some exceeding recommended thresholds. While the authors acknowledge this, the discussion would benefit from a clearer theoretical justification distinguishing these constructs conceptually. A short paragraph explicitly explaining why employability is not reducible to confidence or skill possession alone would improve interpretability.

Interpretation of proficiency as a mediator

The results show that proficiency mediates the technical skills–employability relationship but not the soft skills–employability relationship. This asymmetry is interesting, yet the discussion treats it somewhat briefly. Expanding the explanation, particularly in relation to UI/UX education and curriculum design, would strengthen the contribution.

Gender moderation effects

The moderation results for gender are statistically significant and relevant. However, the manuscript would benefit from a more cautious interpretation, explicitly noting that the study examines perceived employability rather than actual hiring outcomes. This clarification would help avoid overgeneralisation.

Minor language and presentation issues

There are several instances of duplicated phrases, inconsistent spacing, and minor grammatical errors (for example, repeated sentences in the literature review and discussion sections). These do not undermine the study but should be corrected for readability, especially as PLOS ONE does not copyedit accepted manuscripts.

Overall, this is a well-executed study with clear practical implications for IT education and employability development in non-Western contexts. With minor revisions addressing the points above, the manuscript would be suitable for publication.

**Do you want your identity to be public for this peer review?** For information about this choice, including consent withdrawal, please see our For information about this choice, including consent withdrawal, please see our Privacy Policy .

Reviewer #2: No

Reviewer #3: **Yes:** Cheng Kin MengCheng Kin Meng

---

## [Author Response · Author response to Decision Letter 2]

12 Mar 2026

Please refer response to the reviewer's document for the details responce

---

## [Editor Report · Decision Letter 2]

22 Mar 2026

Assessing the Influence of Diverse Skills on Employability Outcomes for IT Undergraduates

PONE-D-25-23960R2

Dear Dr. Wisenthige,

We’re pleased to inform you that your manuscript has been judged scientifically suitable for publication and will be formally accepted for publication once it meets all outstanding technical requirements.

Kind regards,

Rea Lavi

Academic Editor

PLOS One

Additional Editor Comments (optional):

Please make sure that all the images in the manuscript are in high resolution with text that is clearly legible. See Figure 02 for example.
---

## [Editor Report · Acceptance letter]

PONE-D-25-23960R2

PLOS One

Dear Dr. Wisenthige,

I'm pleased to inform you that your manuscript has been deemed suitable for publication in PLOS One. Congratulations! Your manuscript is now being handed over to our production team.

Kind regards,

on behalf of

Dr. Rea Lavi

Academic Editor

PLOS One